# Prolactin and fMRI response to SKF38393 in the baboon

Brad Miller[1], Lauren A. Marks[2], Jonathan M. Koller[1], Blake J. Newman[3], G. Larry Bretthorst[4] and Kevin J. Black[1,3,4,5]

[1] Department of Psychiatry, Washington University School of Medicine, St. Louis, MO, USA
[2] Washington University School of Medicine, St. Louis, MO, USA
[3] Department of Neurology, Washington University School of Medicine, St. Louis, MO, USA
[4] Department of Radiology, Washington University School of Medicine, St. Louis, MO, USA
[5] Department of Anatomy & Neurobiology, Washington University School of Medicine, St. Louis, MO, USA

Corresponding author
Kevin J. Black, kevin@wustl.edu

## ABSTRACT

**Background.** This study's goal was to provide dose–response data for a dopamine agonist in the baboon using standard methods (replicate measurements at each dose, across a range of doses), as a standard against which to subsequently validate a novel pharmacological MRI (phMRI) method. Dependent variables were functional MRI (fMRI) data from brain regions selected *a priori*, and systemic prolactin release. Necessary first steps included estimating the magnitude and time course of prolactin response to anesthesia alone and to various doses of agonist. These first steps ("time course studies") were performed with three agonists, and the results were used to select promising agonists and to guide design details for the single-dose studies needed to generate dose–response curves.

**Methods.** We studied 6 male baboons (*Papio anubis*) under low-dose isoflurane anesthesia after i.m. ketamine. Time course studies charted the changes in plasma prolactin levels over time after anesthesia alone or after an intravenous (i.v.) dose of the dopamine $D_1$-like agonists SKF82958 and SKF38393 or the $D_2$-like agonist pramipexole. In the single-dose dopamine agonist studies, one dose of SKF38393 (ranging from 0.0928–9.28 mg/kg, $N = 5$ animals) or pramipexole (0.00928–0.2 mg/kg, $N = 1$) was given i.v. during a 40-min blood oxygen level dependent (BOLD) fMRI session, to determine BOLD and plasma prolactin responses to different drug concentrations. BOLD response was quantified as the area under the time-signal curve for the first 15 min after the start of the drug infusion, compared to the linearly predicted signal from the baseline data before drug. The $ED_{50}$ (estimated dose that produces 50% of the maximal possible response to drug) for SKF38393 was calculated for the serum prolactin response and for phMRI responses in hypothalamus, pituitary, striatum and midbrain.

**Results.** Prolactin rose 2.4- to 12-fold with anesthesia alone, peaking around 50–90 min after ketamine administration and gradually tapering off but still remaining higher than baseline on isoflurane 3–5 h after ketamine. Baseline prolactin level increased with age. SKF82958 0.1 mg/kg i.v. produced no noticeable change in plasma prolactin concentration. SKF38393 produced a substantial increase in prolactin release that peaked at around 20–30 min and declined to pre-drug levels in about an hour. Pramipexole quickly reduced prolactin levels below baseline, reaching a nadir 2–3 h after infusion. SKF38393 produced clear, dose-responsive

BOLD signal changes, and across the four regions, $ED_{50}$ was estimated at 2.6–8.1 mg/kg.

**Conclusions.** In the baboon, the dopamine $D_1$ receptor agonist SKF38393 produces clear plasma prolactin and phMRI dose–response curves. Variability in age and a modest sample size limit the precision of the conclusions.

## INTRODUCTION

We recently described a new pharmacological MRI (phMRI) method designed to estimate a dose–response curve in a single imaging session (*Black, Koller & Miller, 2013*). We sought to validate that method using specific dopamine receptor agonists in the baboon, a primate species with a large enough brain to allow using functional MRI (fMRI) methods commonly used in humans (*Black et al., 2009*). However, limited data have been available on dose–response curves for fMRI responses to dopamine agonists in primate brain. More data are available on the systemic prolactin response to some dopamine agonists. The goal of the present work was to use a traditional approach to characterize the dose–response curves for regional fMRI signal and prolactin responses in baboon to at least one pharmacologically specific dopamine agonist.

Primate phMRI studies, and before that *ex vivo* and positron emission tomography studies of regional cerebral metabolism and blood flow, have been reported for levodopa and several dopamine agonists (*Black et al., 2000*; *Black et al., 2002*; *Chen et al., 1997*; *Herscovitch, 2001*; *Hershey et al., 2000*; *Hershey et al., 2003*; *Perlmutter, Rowe & Lich, 1993*; *Trugman & Wooten, 1986*; *Trugman & James, 1992*; *Trugman & James, 1993*; *Zhang et al., 2000*). However, rarely have phMRI dose–response curves been reported for drugs of any kind in primates (briefly reviewed in *Black et al., 2010*).

Prolactin is a hormone best known for its ability to promote lactation in response to the suckling stimulus of hungry young mammals, but it plays a role in many biological functions (*Freeman et al., 2000*). Prolactin is produced principally by the anterior pituitary gland, and also within the central nervous system, the immune system, the uterus and the mammary gland. Pituitary prolactin secretion is mediated in large part by dopamine synthesized by the tuberoinfundibular dopaminergic neurons of the arcuate nucleus through $D_2$ receptors (*Freeman et al., 2000*; *Jaber et al., 1996*; *Ben-Jonathan & Hnasko, 2001*). Dopamine is the principal prolactin inhibiting factor and increasing dopamine directly by giving dopamine i.v. reduces plasma prolactin concentration (*Leblanc et al., 1976*; *Norman, Quadri & Spies, 1980*; *Freeman et al., 2000*; *Ben-Jonathan & Hnasko, 2001*). $D_1$ receptor activation also increases prolactin levels most likely by inhibiting the $D_2$

pathway (*Curlewis, Clarke & McNeilly, 1993*; *Colthorpe et al., 1998*; *Durham et al., 1998*; *Freeman et al., 2000*).

SKF82958 is a full agonist at dopamine $D_1$-like receptors, though there is some debate about full vs. partial agonism (*Gilmore et al., 1995*; *Bergman et al., 1996*). SKF82958 produces dose-responsive effects on brain activity (*Black et al., 2000*), but its effects on prolactin secretion are not well described. Pramipexole is a $D_3$-preferring $D_2$-like dopamine agonist (*Mierau, 1995*; *Mierau et al., 1995*; *Piercey, 1998*) that produces regionally specific effects on brain activity (*Black et al., 2002*) and reduces prolactin secretion in humans (*Schilling, Adamus & Palluk, 1992*; *Samuels et al., 2006*). For both of these drugs there is strong evidence that they affect regional cerebral blood flow (rCBF) via changes in neuronal activity and not via direct vascular effects (*Black et al., 2000*; *Black et al., 2002*). In fact, with few exceptions (such as ergot derivatives), synthetic dopamine agonists generally do not uncouple brain blood flow from brain metabolism, i.e., are not expected to affect the BOLD signal except through neuronal effects (discussed in *Black et al., 2000*; *Black et al., 2002*).

SKF38393 is a partial $D_1$ agonist that increases prolactin levels and changes regionally specific brain activity (*Engber et al., 1993*; *Trugman & James, 1992*; *Morelli et al., 1993*). There is little information on fMRI or rCBF effects of SKF38393 (*Dixon et al., 2001*), but it does affect neuronal function and regional cerebral metabolism via dopamine $D_1$ receptors (*Trugman & James, 1993*). SKF38393 has been shown to increase prolactin anywhere from 3- to 40-fold in several species including humans. Its effect on prolactin levels is rapid, peaking about 15–20 min after parenteral administration in the rat or sheep (*Cocchi et al., 1987*; *Tanimoto, Tamminga & Chase, 1987*; *Curlewis, Clarke & McNeilly, 1993*), and prolactin levels return to baseline in about an hour. When given orally, the delay is 60–150 min, returning to baseline within two to three hours (in humans; *Fabbrini et al., 1988*). Some, but not all, studies report dose–response characteristics with regard to prolactin secretion (*Saller & Salama, 1986*; *Cocchi et al., 1987*; *Tanimoto, Tamminga & Chase, 1987*; *Fabbrini et al., 1988*; *Curlewis, Clarke & McNeilly, 1993*; *Durham et al., 1998*).

Here we report preliminary data from male baboons on the timing and magnitude of prolactin response to anesthesia alone ($N = 6$), to SKF82958 ($N = 1$), to SKF38393 ($N = 5$), and to pramipexole ($N = 1$). Then we compare the prolactin and regional BOLD (blood oxygen level dependent) fMRI responses to SKF38393 ($N = 5$).

## MATERIALS & METHODS

### Subjects

The study was approved by the Washington University Animal Studies Committee (protocol # 20090112). We studied six healthy male olive baboons (*Papio anubis*). This species was chosen with a goal of later translation to human studies; specifically, baboon anatomy and neuroendocrine responses are similar to humans', and their brain volume is big enough to allow neuroimaging methods borrowed from human fMRI (*Black et al., 2009*). Three were fully adult (age 6.8–14 years) and three were juveniles (age 4 years). In

captive male baboons, puberty occurs at about 3.5 years of age (*VandeBerg, 2009*, p. xxi), and the testes had descended in all subjects before these studies began.

## Overall approach

We report results from three types of studies: time-course studies with anesthesia alone, time-course studies with a dopamine agonist, and single-dose dopamine agonist studies. To minimize drug sensitization and blood loss, no two studies in a given animal occurred less than two weeks apart.

## Time course studies

### Anesthesia alone

First we charted the effects of ketamine and isoflurane on prolactin secretion over time without administering a dopamine agonist, since both these anesthetic agents increase plasma prolactin levels (*Wickings & Nieschlag, 1980*; *Puri, Puri & Anand Kumar, 1981*; *Crozier et al., 1994*; *Hergovich et al., 2001*; *Rizvi et al., 2001*). We took a blood sample as soon as possible after giving ketamine with the expectation that circulating prolactin concentrations would not change for >20 min after i.m. ketamine (*Rizvi et al., 2001*). The goal was to determine when the effects of ketamine and/or isoflurane on prolactin release would reach an approximate steady state. Blood samples were taken in 4- to 20-min intervals for up to 5 h.

### With dopamine agonist

Time course studies were also done for the dopamine agonists to provide initial information on whether the agonist produced a detectable effect on prolactin release in this model and to determine a reasonable sampling time for the later single-dose studies.

The dopamine agonist was given two to three hours after ketamine, because although prolactin concentration was still elevated at these times as a consequence of the anesthetics, it was significantly reduced from its peak at around 60 min and was starting to level off (see Results: anesthesia alone studies). Blood samples were taken at various intervals before the agonists and at 4- to 20-min intervals afterward for a total of up to seven hours. These studies involved either one dose of the dopamine agonist or three increasing doses, each separated by one hour. The drug was infused i.v. over a five-minute period with a syringe pump (Harvard Model 44). Drug infusion time was reckoned from the end of the five-minute infusion period. Doses for SKF38393 ranged from 0.2 to 5.0 mg/kg, SKF82958 was given at 0.1 mg/kg, and pramipexole was given at 0.2 mg/kg.

## Single-dose dopamine agonist studies

In these studies, a 40-min BOLD fMRI scan began at least 2 h after i.m. ketamine and glycopyrrolate. Sedation from i.m. ketamine is long gone by this time, and a prolactin increase after ketamine in rhesus monkeys had returned to baseline by 60 min (*Quadri, Pierson & Spies, 1978*). A syringe pump delivered the dopamine agonist i.v. to anesthetized subjects at a constant rate beginning at 15 min and ending at 20 min into the scan.

Five animals received varying doses of SKF38393, each dose on a different day. The order of doses was random in each subject. The doses used were 0, 0.0928, 0.2, 0.431,

0.928, 2.0, 4.31, and 9.28 mg/kg; not all subjects received all doses. A sixth animal received pramipexole at doses of 0, 0.00928, 0.0431, 0.0928 and 0.2 mg/kg.

In these single-dose studies, blood samples for prolactin levels were taken just before dopamine agonist administration and again 20–30 min after drug infusion ended (for pramipexole, 60 min after infusion). The sampling time for each animal was based on the results of the time course studies.

## Anesthesia

Animals were fasted overnight before anesthesia. After ketamine 6–12 mg/kg i.m. and glycopyrrolate (0.013–0.017 mg/kg i.m., to reduce secretions), orotracheal intubation was performed via direct laryngeal visualization. Anesthesia was maintained with isoflurane and oxygen beginning 10–30 min after ketamine. An i.v. catheter was placed in one or both legs for drug delivery and withdrawing blood samples. Isoflurane delivery rate was adjusted to maintain the concentration of isoflurane in expired air at the lowest level that would reliably maintain sedation during study sessions. This concentration was measured by a Surgivet V9400 monitor (Smiths Medical, Norwell, MA), and the optimal concentration ranged from 0.95% to 1.45% across animals.

## Drug preparation

SKF82958 (catalog # C130, Sigma-Aldrich, St Louis, MO) and SKF38393 (catalog # D047, Sigma-Aldrich, St Louis, MO) were mixed fresh the day of the study, the pH was adjusted with NaOH, then the solution was filter sterilized (0.22 µm) and stored on ice in a light-tight container until use. SKF82958 was prepared with sterile 0.9% saline at 1.0 mg/ml and the pH adjusted to 6.0–6.5. SKF38393 was prepared with sterile water from 1.0 to 7.3 mg/ml and the pH adjusted to 5.5–6.0. Pramipexole (a gift from Boehringer-Ingelheim Pharmaceuticals, Inc., Ridgefield, CT) was mixed with sterile saline at 1 mg/ml and filter sterilized before use. Some pramipexole solution was kept in aliquots at $-80°C$ for later use.

## Plasma prolactin analysis

Prolactin concentration in plasma or serum was measured via radioimmunoassay (University of Wisconsin National Primate Research Center, Madison, Wisconsin). The inter-assay coefficient of variation was 7.89%, and the intra-assay coefficient of variation was 4.09%.

## MRI sequences and initial processing

All imaging was done on a Siemens Trio 3T scanner (Erlanger, Germany) with boosted gradient coils. To facilitate image registration, an MP-RAGE sequence was used to acquire a T1-weighted structural image of the head, flip angle 7°, echo time (TE) 3.51 ms, inversion time 1000 ms, FOV 192 × 192 mm, 192 slices, and repetition time (TR) 1900 ms; and a 3D turbo spin echo sequence was used to acquire a T2-weighted structural image, flip angle 120°, TE 604 ms, field of view 256 × 256 mm, 192 slices, and TR 3200 ms. Voxel size was approximately $(0.8 mm)^3$ for both structural images.

Functional images were acquired using a blood oxygenation level-dependent (BOLD) contrast-sensitive echo-planar sequence with flip angle 90°, TE 30 ms, field of view $384 \times 384$ mm, 32 slices, voxel size $(2.5 \text{ mm})^3$, and TR 2160 ms, with interleaved slice acquisition. Over a period of 40 min, 1111 volumes (frames) were acquired; the first 4 frames were discarded to ensure steady-state magnetization.

Functional images from each subject were preprocessed to reduce artifacts, including correction for intensity differences due to interleaved acquisition, interpolation for slice time correction, correction for head movement, and alignment to atlas space (*Black et al., 2001*). Image intensity was not adjusted on a frame-by-frame basis, but the entire 40-min acquisition was scaled linearly to a mean (across frames) whole-brain modal value of 1000.

## Volumes of interest

Volumes of interest (VOIs) were defined by hand on each animal's MP-RAGE (structural image) for the following regions of brain: hypothalamus, pituitary, striatum, and midbrain. The hypothalamus and pituitary were chosen because of the potential relevance to prolactin release, and the striatum and midbrain to provide comparisons to prior blood flow, metabolic, and BOLD pharmacological activation studies of dopamine agonists. The regions were drawn with reference to *Davis & Huffman (1968)* by one author (BM) with over 20 years' experience in neuroanatomy (including delineating subnuclei in rodent thalamus and tract tracing over the entire length of the primate brain). All regions were drawn before any regional time-signal curves were examined. The VOIs were transferred to the MRI by a validated cross-modality image registration method (*Black et al., 2001*).

## Determination of ED$_{50}$

The mean prolactin plasma concentration response to each dose of SKF38393 in the single-dose studies was plotted against the logarithm of the dose. A sigmoid $E_{max}$ model (*Holford & Sheiner, 1982*) was fit to the data using a Markov chain Monte Carlo (McMC) method (*Bretthorst, 1988*; *Bretthorst & Marutyan, 2012*). To improve sampling across the logarithmic abscissa, a variable $Q = \log_{10}(\text{ED}_{50})$ was defined and the McMC method was used to estimate parameters in the equation

$$E(D) = E_0 + E_{max} \cdot D^n / (D^n + 10^{nQ}),$$

where $D$ = dose of SKF38393 in mg/kg, $E_0$ = response at baseline ($D = 0$), $E(D)$ = response (effect) at dose $D$, and $n$ is the Hill coefficient. The response at each dose was defined as post-drug minus pre-drug prolactin concentration on that same day (hence the nonzero baseline response $E_0$).

Separate dose–response curves and ED$_{50}$ estimates were derived from mean BOLD (fMRI) responses to SKF38393 and pramipexole in each VOI. The BOLD response at a given dose was defined as the area under the BOLD signal curve (AUC) from 15 to 30 min after drug, after accounting for the linear trend in the first 15 min of data:

$$\text{AUC} = \int_{15}^{30} \text{BOLD}(t) - (mt + b) dt,$$

**Table 1** Baseline prolactin plasma concentration, and peak increase in prolactin with anesthesia alone, by age.

| Subject | Age at start of study (years) | Baseline prolactin concentration (ng/ml), mean ± S.D. | Increase in prolactin concentration with anesthesia |
|---------|-------------------------------|-------------------------------------------------------|-----------------------------------------------------|
| R | 4.08 | 2.62 ± 0.35 | 354% |
| T | 4.25 | 3.04 ± 0.41 | 334% |
| B | 4.33 | 2.64 ± 0.34 | 236% |
| E | 6.75 | 4.05 ± 0.94 | 473% |
| S | 6.92 | 3.50 ± 0.96 | 1198% |
| D | 14.08 | 23.5 ± 6.02 | 410% |

where time $t$ is in minutes, BOLD($t$) is the BOLD signal at each time point, and $y = mt + b$ is the least-squares line fit to the first 15 min of BOLD signal data.

# RESULTS

## Baseline prolactin levels

Baseline prolactin levels were measured from the blood sample taken as soon as possible after the subject was given ketamine. The delay between ketamine administration and the first blood sample ranged from 6 to 24 min. These data showed no clear change in prolactin levels during this time period. For example, one subject's first sample was taken as soon as eight minutes or as late as 24 min after ketamine (a total of 11 baseline samples were examined for this subject). Prolactin levels were similar across these time points (data not shown). The other subjects showed a similar pattern. Thus, changes in plasma prolactin levels due to anesthetics appear to occur only after this time period.

Baseline levels of plasma prolactin were positively correlated with age ($r = 0.96$, $p < 0.01$; Table 1). The oldest animal, at 14 years of age, had five-fold higher baseline levels of prolactin than any of the other subjects, but the correlation with age remained significant even excluding this subject. The three juveniles had the lowest baseline measurements.

## Effect of anesthesia alone on prolactin release

Anesthesia increased prolactin in all 6 animals ($p = .03$, 2-tailed binomial distribution), with a peak concentration from 2.4-fold to 12-fold higher than at baseline ($p < .07$, 2-tailed paired t test). The three juveniles had the lowest increase (Table 1).

Prolactin levels peaked between 50 and 90 min after ketamine. Prolactin levels declined partially and leveled off somewhat around 100–120 min (Fig. 1). They continued to decline but remained well above baseline levels for 3–5 h (last measurement, median 208 min) in all animals ($p = .03$, 2-tailed binomial distribution; $p < .07$, 2-tailed paired t test). Some, but not all, of the time course profiles appeared to show a small, secondary increase in prolactin after 100 min. A similar post-decline surge has been recorded in other studies with ketamine (*Rizvi et al., 2001*) and isoflurane (*Reber et al., 1998*).

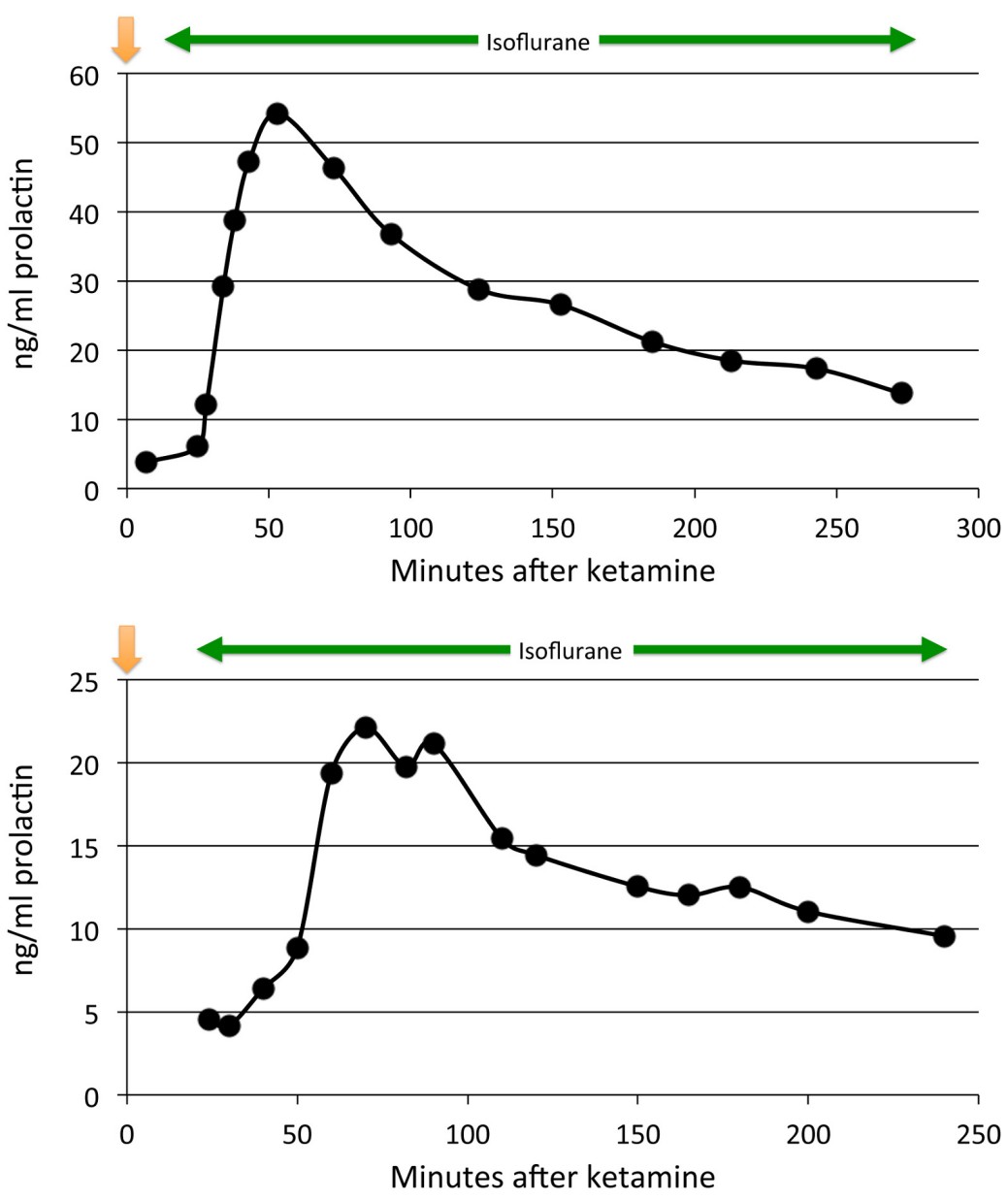

**Figure 1  Plasma prolactin levels over time with anesthesia alone.** Two time course studies of the effect of anesthesia alone on prolactin plasma concentrations in two adults. The green line shows the period of isoflurane inhalation. The orange arrow marks when the ketamine was given.

Prolactin release with anesthesia was greater in the adults than in the juveniles (Fig. 2; $p = 0.045$).

## Time course studies with dopamine agonists

Subsequent experiments attempted to detect an effect from the experimental drug on prolactin release, and to define an appropriate time for sampling. SKF82958 at 0.1 mg/kg

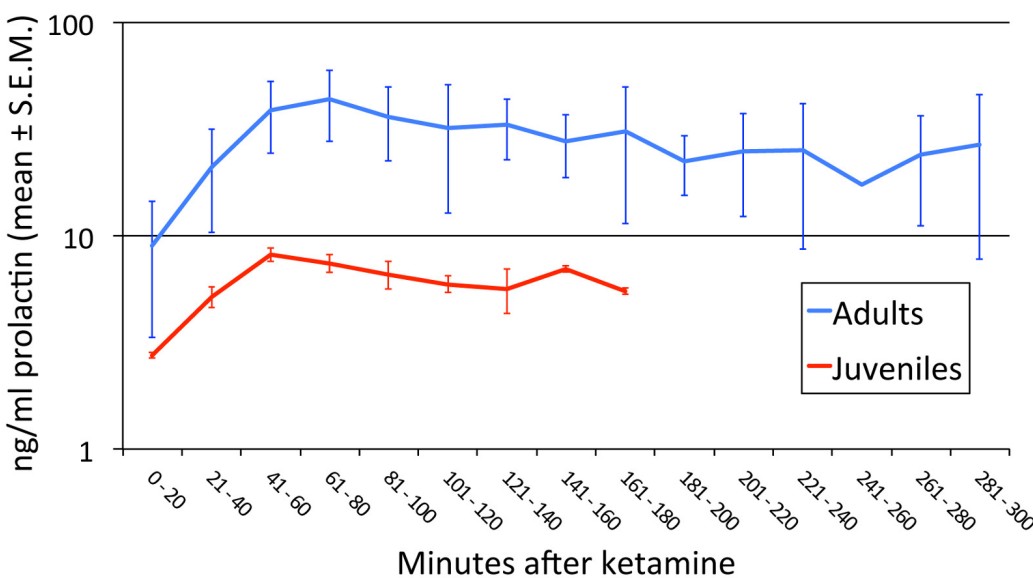

**Figure 2 Age-related differences in prolactin secretion with anesthesia.** We performed an ANOVA with age group as a between-subject factor and time (in 20-min bins) as a repeated measure. Every animal had data out to at least the 121–140 min bin. Three missing values were imputed by averaging the previous and following levels in that animal (these two values differed by only 24%, 13%, and 0%), and in one animal the first sample was at 21′ and was treated as if it had been drawn at 20′. There was a significant difference between juveniles and adults in the [PRL] response to anesthesia (age group × time interaction $F_{6,24} = 7.350$, $p = .045^*$; main effect of age group $F_{1,4} = 4.103$, $p = .113$; main effect of time $F_{6,24} = 12.799$, $p = .018^*$). * = Greenhouse-Geisser correction for nonsphericity.

produced no apparent change in prolactin levels ($N = 3$), which continued to decline over time in a pattern that was indistinguishable from the anesthesia-only condition (Fig. 3).

By contrast, the partial agonist SKF38393 appeared to produce a substantial increase in prolactin (Fig. 4), including in the same adult whose non-response to SKF82958 appears in Fig. 3. Prolactin levels increased as soon as five minutes after the drug infusion, peaked within 20–30 min, then declined more gradually. The peak time was unique for each animal but was 20–30 min after drug; this time determined when that animal's blood sample was to be taken after drug infusion during the single dose studies.

Prolactin declined shortly after pramipexole administration in the single subject tested, with a substantial decline by 60 min and reaching a nadir about three hours later (Fig. 5). Note that the prolactin concentrations dropped to well below baseline level.

## Single-dose studies

### SKF38393 and prolactin

The single dose studies showed a dose-related effect of SKF38393 on prolactin (Fig. 6). Two adults (S, E) showed the most consistent and largest increases in prolactin, resulting in over a 5-fold increase with the larger doses compared to pre-drug levels. One adult (D) and the two juveniles had more modest and inconsistent increases, not exceeding a 2-fold increase over pre-drug levels.

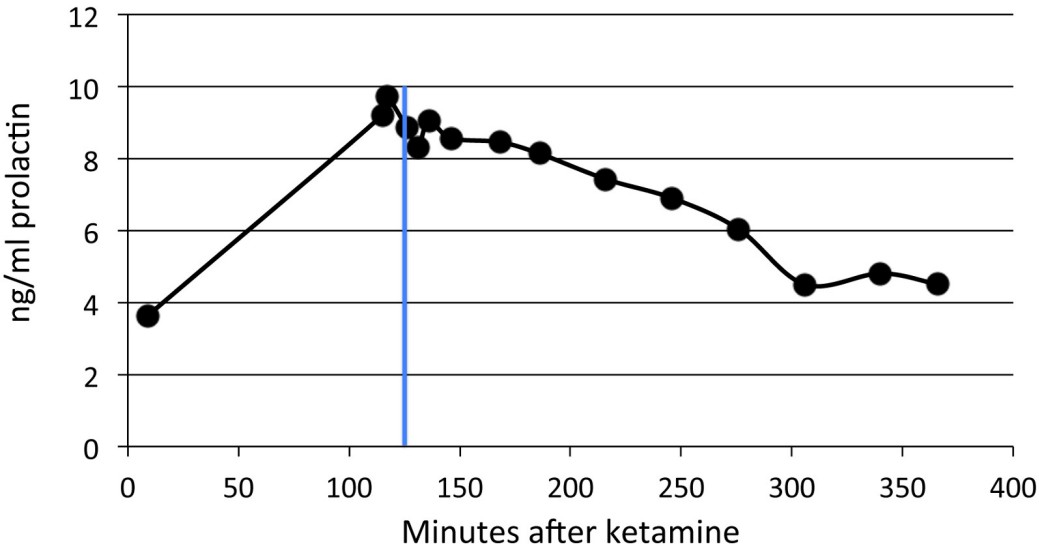

**Figure 3 SKF82958 effects on plasma prolactin.** Data from one of the adult animals. A single dose of 0.1 mg/kg was given at 125 min after ketamine (blue line).

The [PRL] increase with drug was not monotonic. Of course, this comparison involves results obtained on different days, and these results differed from the time course data, in which a clear dose–response effect was visible when different doses were given within the same session (Fig. 4). Figure 6 plots results as the difference in prolactin concentration from the pre-drug levels, but similar variability was seen when the values are plotted as percent of pre-drug levels or absolute concentrations of prolactin.

The group median dose–response curve for prolactin release following BOLD decreased at the highest dose, so strictly the sigmoid $E_{max}$ model may not apply. However, recognizing this limitation, the model fit well to the remaining data points (0–4.31 mg/kg) with an $ED_{50}$ of 1.30 mg/kg ($E_0 = -3.65, E_{max} = 67.38, n = 1.03$; Fig. 7).

### SKF38393 and BOLD

There was a clear effect of SKF38393 on BOLD signal in midbrain at the higher doses (Fig. 8; Fig. 9). BOLD signal in pituitary decreased with increasing dose, but it increased in the other three regions (Fig. 10). The regional responses to the agonist, by dose, were best fit to the sigmoid $E_{max}$ model with $ED_{50}$s of: hypothalamus, 2.57 mg/kg; pituitary, 3.85 mg/kg; midbrain, 5.59 mg/kg; and striatum, 8.06 mg/kg.

### Pramipexole

The one animal given pramipexole (subject T, juvenile) showed a decrease in prolactin concentration with the agonist at all doses tested (Table 2). The BOLD responses from this animal are also given in Table 2.

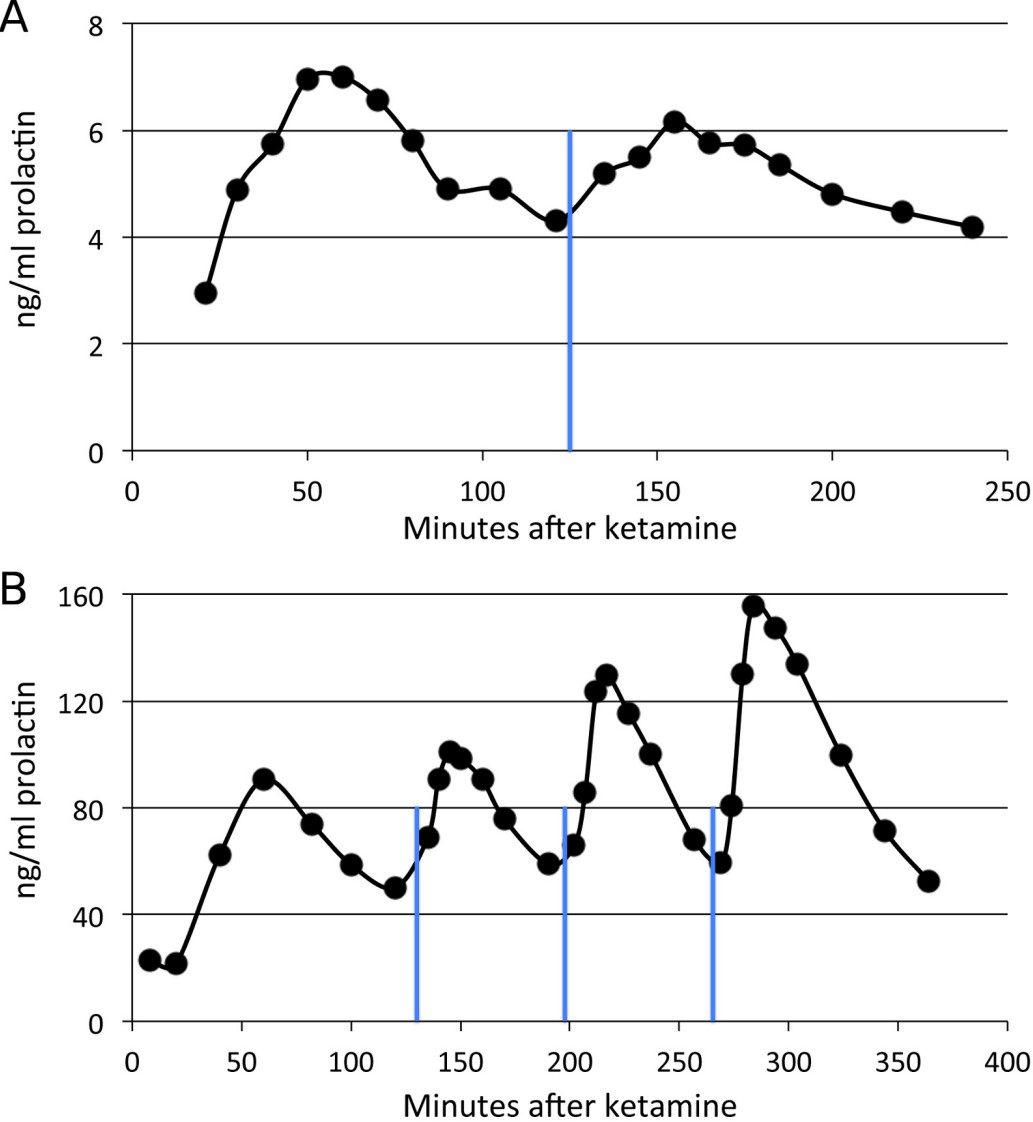

**Figure 4 Time course of SKF38393 effects on plasma prolactin.** (A) Plasma prolactin concentration over time in an animal given a single dose of 2.0 mg/kg of SKF38393 125 min after ketamine. (B) Plasma prolactin concentration over time in another animal given successive doses of 1.0, 2.0 and 5.0 mg/kg of SKF38393 at 130, 197 and 264 min after ketamine. Vertical blue lines show the timing of drug administration.

## DISCUSSION

### Anesthetic effects on prolactin

Our results with the anesthetics ketamine and isoflurane are very similar to those of previous studies, showing an initial rise in prolactin levels followed by a gradual decline (*Reber et al., 1998*; *Puri, Puri & Anand Kumar, 1981*; *Rizvi et al., 2001*; *Hergovich et al., 2001*). Prior studies suggest that isoflurane may produce a larger prolactin response than ketamine, though the data are from different species and sex. Isoflurane in human females

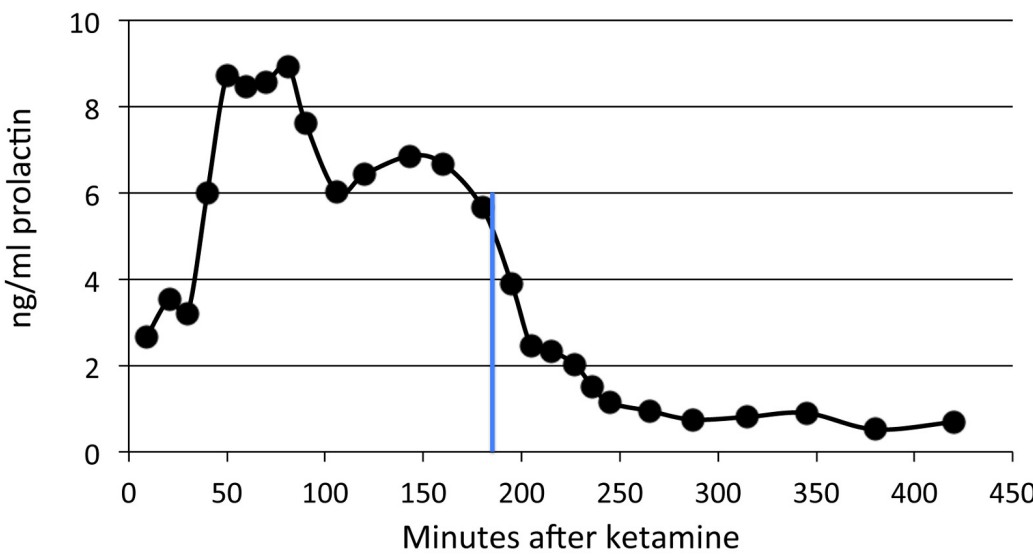

**Figure 5 Time course of pramipexole effects on plasma prolactin.** A single dose of 0.2 mg/kg of pramipexole (marked by the blue line) was given 180 min after ketamine.

**Table 2 Responses to pramipexole by dose (N = 1).** AUC is defined in the last paragraph of Methods.

| Dose (mg/kg) | Change in prolactin plasma concentration (ng/ml) | BOLD signal response, midbrain (AUC) | BOLD signal response, pituitary (AUC) | BOLD signal response, hypothalamus (AUC) | BOLD signal response, striatum (AUC) |
|---|---|---|---|---|---|
| 0 | −0.62[a] | [b] | [b] | [b] | [b] |
| 0.00928 | −2.73 | 130.5 | 15.9 | 115.1 | 140.2 |
| 0.04310 | −8.10 | 54.0 | −92.4 | −22.6 | 60.6 |
| 0.09280 | −3.53 | 47.3 | −80.2 | 121.2 | 73.3 |
| 0.20000 | −6.04 | 187.0 | −92.7 | −85.2 | 54.2 |

**Notes.**

[a] The prolactin response after the saline control is nonzero because it is defined as post-minus pre-drug, and prolactin levels are gradually declining following their initial post-ketamine peak (see Fig. 1).

[b] Not available (marked artifact in numerous frames during this run).

has shown a 9 to 12-fold increase in prolactin (*Reber et al., 1998*; *Marana et al., 2003*), while ketamine produced a 2- to 4-fold increase in male macaques after a single injection (*Puri, Puri & Anand Kumar, 1981*; *Rizvi et al., 2001*). A study in adult baboon females found no significant difference in PRL release between constant-rate i.v. ketamine and inhaled halothane, both beginning after i.m. ketamine (*Walker, 1987*).

However, we saw a substantial range in the magnitude of the increase in our baboons, with the lowest being 2.4-fold and the highest 12-fold. Five of the six subjects had an increase of prolactin from anesthetics alone that ranged from 2.4-fold to 4.7-fold compared to baseline values, suggesting that isoflurane was not producing as dramatic an effect in our male baboons as it does in human females. In this study, we cannot reliably separate the effects of the two anesthetics, but since ketamine's anesthetic effects are largely gone an hour after i.m. administration (discussed in *Black, Gado & Perlmutter, 1997*), the

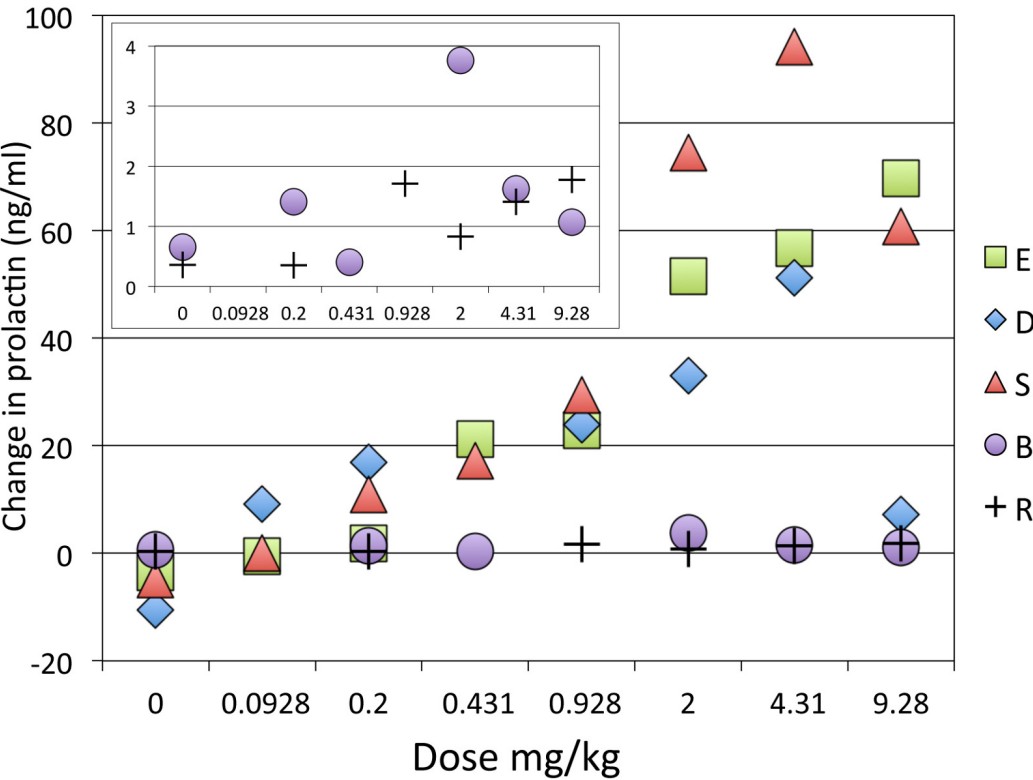

**Figure 6 Dose-response curve for effect of SKF38393 on prolactin.** Prolactin plasma concentration after SKF38393 is plotted as the difference from each day's pre-drug concentration. The inset shows a close-up of the data from the two juveniles. Adults and juveniles differed in their responses (repeated measures ANOVA, main effect of age group $F_{1,3} = 18.068$, $p = .024$; dose × age group interaction $F_{7,21} = 4.780, p = .077*$; main effect of dose $F_{7,21} = 5.154, p = .068*$. * = Greenhouse-Geisser correction for nonsphericity.

prolactin effects after 2–5 h likely are primarily attributable to isoflurane. However, even this conclusion must be tempered by the caveat that drug effects on prolactin release are somewhat delayed.

The blood samples taken as soon as practical after ketamine administration suggested that prolactin levels did not begin to increase significantly until >20 min after ketamine, though we necessarily lacked pre-ketamine data for comparison.

## Dopamine agonist effects on prolactin

Our time course results with SKF38393 are similar to those previously published, with a rapid increase in prolactin levels after drug administration (*Tanimoto, Tamminga & Chase, 1987*). In some instances we observed an increase in as little as five minutes after drug infusion. Prolactin levels peaked by 20 to 30 min, followed by a gradual decline (*Tanimoto, Tamminga & Chase, 1987*; *Curlewis, Clarke & McNeilly, 1993*; *Cocchi et al., 1987*).

Prolactin responses generally increased with higher doses of SKF38393, but in 3 animals the greatest response occurred at 2 or 4.31 mg/kg rather than at the highest dose, 9.28 mg/kg. This may simply be due to variability in the single-dose data as discussed

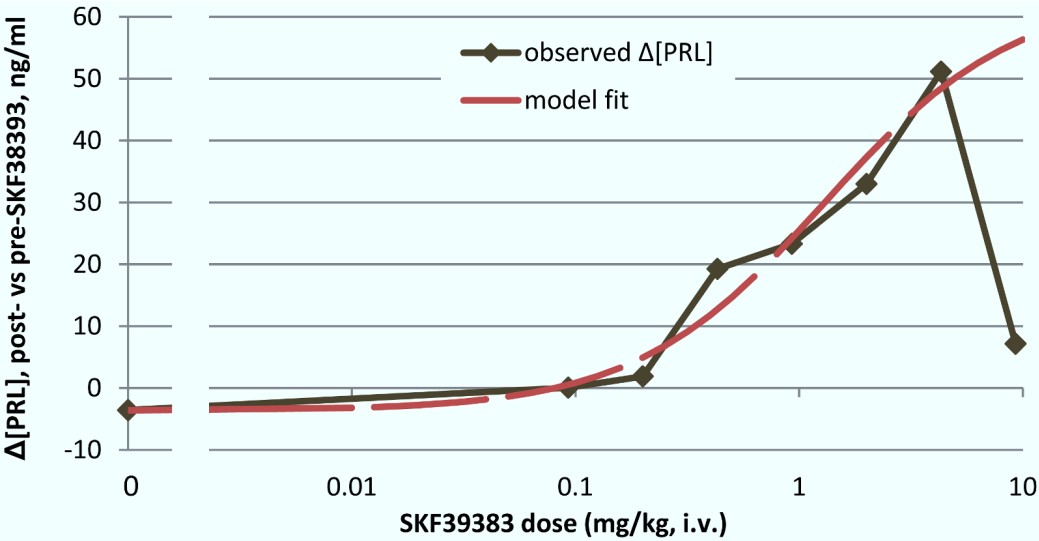

**Figure 7 ED₅₀ for the mean prolactin response to SKF38393.** The mean plasma [PRL] response to SKF38393 in the single-dose studies was plotted against log(dose). The response at each dose was defined as post-drug minus pre-drug [PRL] (hence the nonzero "response" in the absence of drug). A sigmoid $E_{max}$ model was fit to the data from 0 to 4.31 mg/kg as described in Materials & Methods under "Determination of ED₅₀".

below. Alternatively, the physiological response of prolactin release may have an inverted U shape. Resolving this question would require higher doses of drug and more subjects. We ignored the data from the highest point in order to estimate an ED₅₀ for prolactin response, but we recognize the limitations in this approach.

The only previous data we found for prolactin response to the $D_1$ agonist SKF82958 was from an *in vitro* study (*Burris, Stringer & Freeman, 1991*). Based on its action on $D_1$ receptors and previous data from SKF38393, we predicted that SKF82958 would stimulate prolactin release. At the dose used in the present study, SKF82958 produces changes in regional cerebral blood flow (*Black et al., 2000*), but there was no clear evidence of a change in prolactin release after this dose. One speculative explanation might be that $D_1$ full agonists do not affect prolactin release, while the partial agonist SKF38393 increases prolactin release by its relative antagonism in the setting of intrinsic dopamine release.

The results from our pramipexole data are similar to those reported for human studies. We see a dose-dependent reduction in prolactin levels (*Schilling, Adamus & Palluk, 1992*) lasting for four hours or more after a single dose, with the lowest values seen several hours after administration (*Schilling, Adamus & Palluk, 1992*; *Porta et al., 2009*; *Hood et al., 2010*).

## Variability in the single-dose studies of prolactin response to SKF38393

Our time-course studies suggested a clear dose–response effect from SKF38393 on plasma prolactin levels (Fig. 4B), in that more drug produced more prolactin. Most other studies report a similar effect (*Tanimoto, Tamminga & Chase, 1987*; *Fabbrini et al., 1988*; *Curlewis, Clarke & McNeilly, 1993*; *Cocchi et al., 1987*; *Saller & Salama, 1986*; *Durham et al., 1998*).

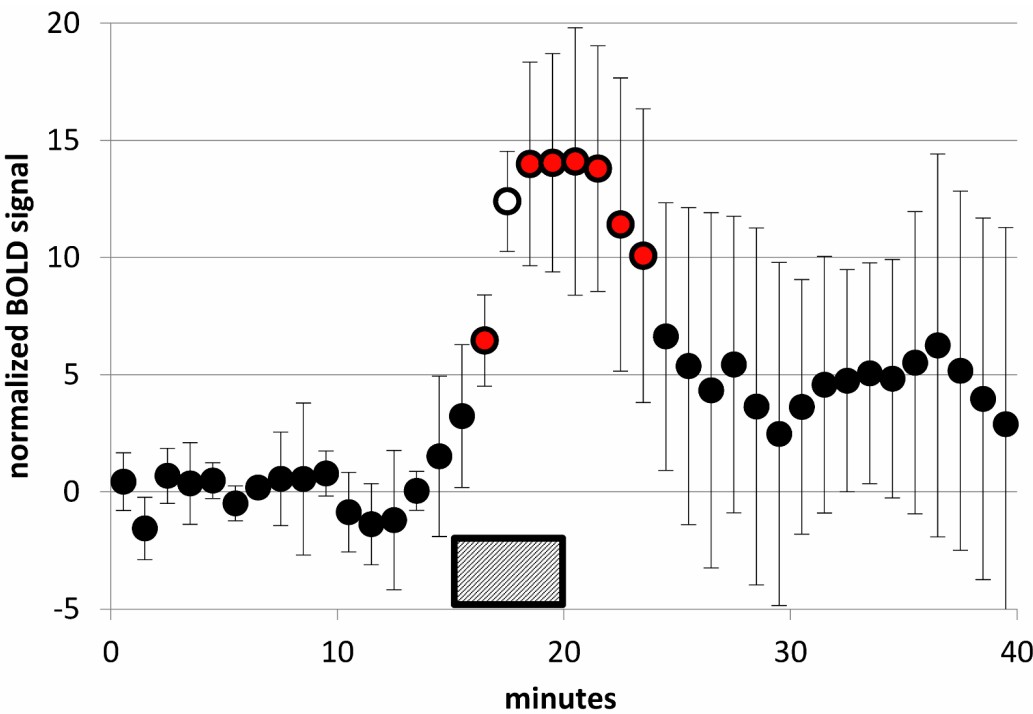

**Figure 8 SKF38393 2 mg/kg caused a clear BOLD signal response in midbrain.** BOLD signal in the midbrain VOI during a 40-min functional MRI scan, after subtracting the linear trend computed from the first 15 min of data. Data plotted here are from each animal's 2 mg/kg day from the SKF38393 single-dose studies (mean $\pm$ S.D., $N = 5$). SKF38393 2 mg/kg was given i.v. over the time interval indicated (15–20 min). Markers filled with red indicate points that differ significantly from zero ($p < 0.05$, 2-tailed single-sample $t$ test); the marker filled with white indicates $p = 0.0002$.

However, as noted above, this relationship was less pronounced in the single-dose studies, and possible reasons for this include the following.

The single-dose studies were conducted on separate days, and the physiological response could vary for that reason, even though they were expressed in relation to the pre-drug levels. The pre-drug levels varied from one session to another by as much as four-fold for a single animal. This variability was present despite our efforts to maintain the expired isoflurane at a consistent level throughout the study. With the small number of subjects studied, deviations from a monotonic increase may reflect merely noise in the data.

Another potential source of variability is that a single post-drug sample, even at a consistent time, may not always catch the maximal effect of the drug on prolactin concentration. Also, though we attempted to maintain a similar time schedule (a two hour delay from the time we first gave ketamine to the start of the BOLD scan), that varied somewhat due to factors not under our control. Thus, there are multiple factors that could contribute to the variability we saw in the single-dose data.

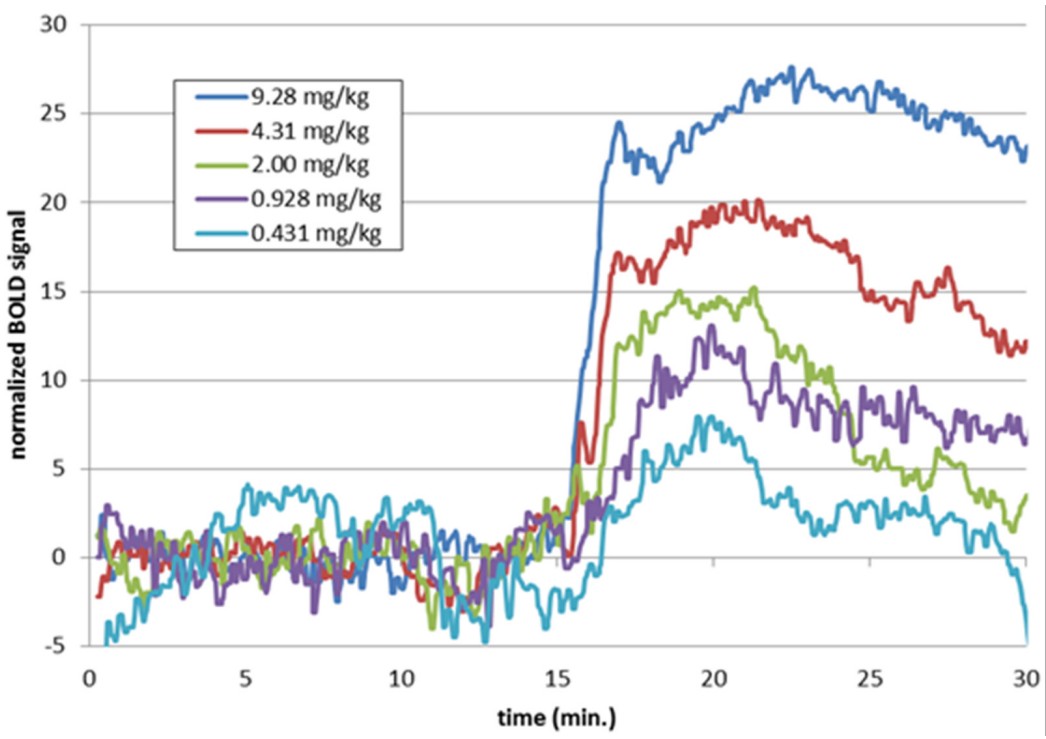

**Figure 9 Time-BOLD curves for SKF38393 in midbrain, by dose.** Mean BOLD signal over time in the midbrain VOI for the five highest doses of SKF38393 ($N = 5$). Each signal curve was detrended as described in the previous figure legend. The drug was given i.v. between the 15 and 20 min marks.

## Age effects on prolactin

Our data suggest that baboons may have age-related differences in basal plasma prolactin levels. Our sample size is too small to reach a firm conclusion in this respect. However, recently presented data from a large group of male *P. anubis* baboons showed that serum prolactin does increase with age, peaking at around 13 years; fully adult males have significantly higher serum prolactin concentrations than four-year-old males (*Phillips-Conroy et al., 2013*). A report by *Kondo et al. (2000)* showed that prolactin levels increase about 3-fold (on average) with age in male chimpanzees. However, this report does not specify how long the blood sample was taken after administering ketamine and the dopamine $D_2$-like antagonist droperidol, and both of these agents increase prolactin (*Puri, Puri & Anand Kumar, 1981*; *Hayashi & Tadokoro, 1984*; *Schettini et al., 1989*; *Rizvi et al., 2001*; *Hergovich et al., 2001*). For comparison, human males have no significant differences in prolactin levels from prepubescence to adulthood (*Guyda & Friesen, 1973*; *Aubert, Grumbach & Kaplan, 1974*; *Lee, Jaffe & Midgley, 1974*; *Ehara, Yen & Siler, 1975*; *Finkelstein et al., 1978*).

The fully adult animals also showed a greater prolactin response to the anesthetics and to SKF38393 (Table 1, Figs. 2 and 6).

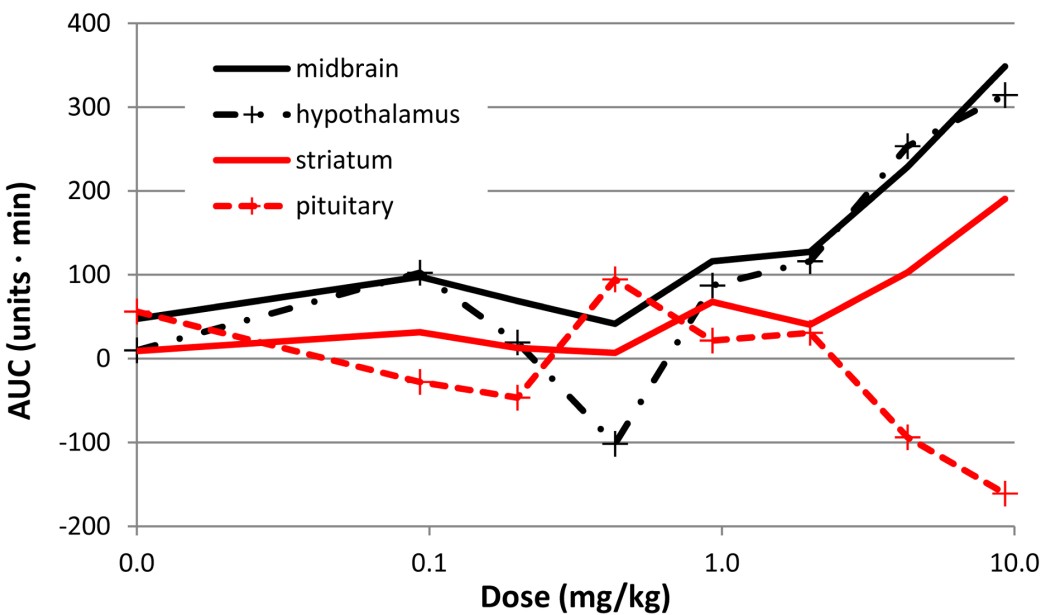

**Figure 10** **Dose-response curves for the effect of SKF38393 on BOLD signal.** The AUC at each dose of SKF38393 is plotted (mean, $N = 5$) for midbrain, hypothalamus, pituitary, and striatum. AUC is defined in the last paragraph of Methods.

## BOLD responses to SKF38393

Noting the caveats above, the estimated prolactin $ED_{50}$ for SKF38393 was 1.30 mg/kg. It seems relevant that the VOI with the closest BOLD $ED_{50}$ value to this was hypothalamus, with an $ED_{50}$ of 2.57 mg/kg.

## CONCLUSIONS

In the baboon, the $D_1$ agonist SKF38393 increased plasma prolactin levels, and caused dose-related changes in BOLD signal in several brain regions, hypothalamus, pituitary and midbrain being more sensitive than striatum. From limited data, the $D_2$-like agonist pramipexole produced a decline in prolactin levels, and the $D_1$ agonist SKF82958 did not affect prolactin release.

## ACKNOWLEDGEMENTS

We thank Dr. Jane Phillips-Conroy for helping us assess age and sexual maturity in baboons, and for sharing with us her recent data on prolactin in baboons.

### Funding

Funding was provided by the U.S. National Institutes of Health (1R21MH081080 and ARRA supplement 3R21MH081080-01A1S1; K24 MH087913; T35 HL007815) and the McDonnell Center for Systems Neuroscience at Washington University. Boehringer-Ingelheim graciously provided the pramipexole. Boehringer-Ingelheim required that we

administer no single dose higher than 0.2 mg/kg i.v., but otherwise did not affect the study design, data collection, or analysis, or the writing of the manuscript.

## Grant Disclosures
The following grant information was disclosed by the authors:
U.S. National Institutes of Health: 1R21MH081080, 3R21MH081080-01A1S1, K24MH087913, T35HL007815.

## Competing Interests
Kevin J. Black is an Academic Editor for PeerJ.

## Author Contributions
- Brad Miller conceived and designed the experiments, performed the experiments, analyzed the data, wrote the paper.
- Lauren A. Marks, Jonathan M. Koller and Blake J. Newman analyzed the data, reviewed and edited the manuscript.
- G. Larry Bretthorst contributed reagents/materials/analysis tools, reviewed and edited the manuscript.
- Kevin J. Black conceived and designed the experiments, analyzed the data, wrote the paper.

## Animal Ethics
The following information was supplied relating to ethical approvals (i.e., approving body and any reference numbers):

Washington University Animal Studies Committee, protocol # 20090112.

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
