# Peer review of "Prolactin and fMRI response to SKF38393 in the baboon"

_PeerJ, doi:10.7717/peerj.195_

## Round 0.1 · original submission · Major Revisions

Dear Authors, Please pay close attention to the comments of these two peer reviewers, who will re-review the revised manuscript.

Reviewer 1 ·

Basic reporting

The article is fairly well written and a clear hypothesis put forward. The methodology is fairly standard and appropriate although authors should indicate the inter- and intra-assay coefficients of variation for the prolactin RIA. The discussion is clear and actually focused and to the point.

Experimental design

The experimental design is satisfactory and fairly well presented.

Validity of the findings

The findings are described adequately but the validity is not supported by statistical evaluation. The authors needs to perform statistics in order to verify comments made in the results and discussion. The figure legends are incomplete and should include the statistical tests applied and number of animals studied to achieve that. There are way too many figures and the authors need to reduce this for example by placing some of the data in tables (e.g. figs 1 and 2) or combine some of the figures into 2-4 panel components. The statements about age are lacking statistical support. The authors should indicate whether the 4 year old animals were prepubertal or postpubertal. The single aged male may have a higher basal prolactin but that may reflect differences in body weight, sensitivity to ketamine-for example, if this animal was previously studied and injected with ketamine-he may have needed more ketamine for this study. Authors should comment. Also, dose of ketamine/kg body weight should be indicated and whether animals were fasted overnight prior to study. Finally the data in figure 4 looks different from that in figure 3 and perhaps reflects presentation of overall means in fig 4 vs 3 but authors should comment.

Additional comments

The authors have done a good job with references although a previous study (Am J Primatology13:325-332, 1987) in adult female baboons (Papio anubis) could prove be helpful in explaining some of the queries posed by the authors, e.g. in the discussion and in the results, p 5 ,lines 20-26.

·

Basic reporting

No comments

Experimental design

No comments

Validity of the findings

No comments

Additional comments

The authors present a manuscript describing the timecourse of prolactin and phMRI response to dopamine agonists in the baboon brain

Overall, the manuscript lacks details in many places and requires clarification of the study methods and conclusions.

The introduction could be much clearer on why this study is interesting, why it is important, why it was conducted in baboons and what role fMRI plays. Indeed the background section of the abstract does not mention fMRI, suggesting it is unrelated to the overall aims of the study. The introduction introduces prolactin as a hormone and its control by dopamine; then SKF38393 is introduced as a specific compound that increases prolactin and SKF82958 is described as a compound whose effects on prolactin are not known; pramipexole is introduced as a compound that reduces prolactin secretion and also has known effects on brain activity. The authors then state the experiments they have conducted and what data was collected. Some of this data is described as tentative (n=1), but no clear reasoning of the purpose of the work is given and no hypotheses are presented. Thus the introduction falls short of its purpose and I recommend it is rewritten, bearing in mind the preceding comments.

The choice of baboons is not sufficiently justified. Why was this done in baboons and not rats or humans? I suspect for the latter the authors may not receive permission to administer the specific drugs. But the “reasons related to brain imaging,” is vague and should be expanded upon. Why mix adults and juveniles?

On page 2 please add the number of animals to the section on anesthesia alone.

On page 4 please add details of the imaging slice sequence (interleaved or sequential top-down/bottom-up. This information is linked to the application of slice time correction. Also, information regarding when anesthesia was initiated in relation to the scanning acquisition was not apparent.

Please provide more information on the intensity normalisation. Given that BOLD is not quantitative and the signal is affected by low frequency fluctuations the concern is that intensity normalisation per image may be inappropriate.

For the determination of ED50 please consider providing more information in the figure legends so that are ‘stand-alone.’ For example, please avoid providing a figure legend which generally refers the reader to the text. The basic information added in to the legend will make the manuscript easier to read, especially considering the number of figures. It is mentioned on page 4 that dose-response curves and ED50 were derived for each ROI. However, the figures only seem to show the midbrain. Could the authors consider adding the other regions to figure 12 so that the data for different regions can be examined by the reader?

For the VOI definition it is great that they were defined for each individual animal on their structural image. What is not clear is the protocol used for these definitions and the quality of their implementation (e.g. inter/intra-rater reliability). Please provide details or references.

The effects of anesthesia alone on BOLD signal does not appear to be given. This is very important as ketamine is also a potent dopamine D2 receptor antagonist.
In the discussion, there is little mention of the literature trying to unpick the effect of dopamine drugs on the BOLD signal. In particular the degree to which the effects on BOLD can be explained by direct effects on the vasculature, versus the effects on neural activity.

Overall, there is not a clear sense as to why the results are important. What is the biological relevance? What is the translational value?

Unfortunately there is too much anecdotal commentary in the discussion too, due to the design – the effects on age (n=3 vs 2), the effects of pramipexole (n=1). The information on the effects of SKF38393, including the dose response is the clearest in terms of findings and rightly forms the focus of the discussion.

---

## Round 0.2 · accepted · Accept

The two peer reviewers were unable to re-review the revised manuscript.As the Academic editor I reviewed the revised manuscript in accordance to the major changes made including Figures/Tables and reread all the associated references and found that the revisions were made accordingly.I therefore have allowed the manuscript to be accepted in it's revised format.Thank you for the hard work done to revise it as suggested by the peer reviewers.Let us hope the medical community can use this data to help forward the results of this experiment to another translational level.